# Adhesion to a CAD/CAM Composite: Causal Factors for a Reliable Long-Term Bond

**DOI:** 10.3390/jfb13040217

**Published:** 2022-11-03

**Authors:** Sandra M. Duma, Nicoleta Ilie

**Affiliations:** Department of Conservative Dentistry and Periodontology, University Hospital, Ludwig Maximilians University (LMU), 80336 Munich, Germany

**Keywords:** CAD/CAM resin-based composite, luting materials, shear bond strength, sandblasting, bonding

## Abstract

Computer aided design/manufacturing (CAD/CAM) technology has become an increasingly popular part of dentistry, which today also includes CAD/CAM resin-based composite (RBC) applications. Because CAD/CAM RBCs are much more difficult to bond, many methods and attachment materials are still being proposed, while the best application method is still a matter of debate. The present study therefore evaluates causal factors for a reliable long-term bond, which includes the surface preparation of the CAD/CAM RBC, aging and the type of luting material. The reliability of the bond was calculated, and supplemented by fractography to identify fracture mechanisms. Five categories of luting materials were used: (1) temporary zinc phosphate cement, (2) glass ionomer cement (GIC), (3) resin-modified GIC, (4) conventional adhesive resin cement (ARC), and (5) self-adhesive RC. Half of the CAD/CAM RBC surfaces (*n* = 200) were sandblasted (SB) with 50 µm aluminum oxide, while the other half remained untreated. Bond strength measurements of the 400 resulting specimens were carried out after 24 h (*n* = 200) or after additional aging (10,000 thermo-cycles between 5 and 55 °C) (*n* = 200). The data were statistically analyzed using one- and three-way ANOVA followed by Games-Howell post-hoc test (α = 0.05) and Weibull analysis. Aging resulted in a significant decrease in bond strength primarily for the conventional cements. The highest bond strengths and reliabilities were recorded for both ARCs. SB caused a significant increase in bond strength for most luting materials, but also caused microcracks in the CAD/CAM RBC. These microcracks might compromise the long-term reliability of the bond in vivo.

## 1. Introduction

An increased interest in computer-aided design/manufacturing (CAD/CAM) prosthetic restoration materials, such as glass ceramics, polymer-infiltrated ceramics, and resin-based composites (RBC), has emerged in the recent years [1,2]. The latter came into view as industrial heat-polymerization under high pressure was applied to conventional RBCs, which led to significant improvements in the mechanical characteristics of CAD/CAM RBCs [3,4]. But clinical success is based on the longevity of the restoration and thus, also on bonding reliability [5]. Due to the high degree of conversion (DC) during fabrication of CAD/CAM RBCs [4], the formation of a chemical bond is unlikely. As a result, roughening of the surface by airborne-particle abrasion, hereafter referred to as sandblasting (SB), has been established to enable micromechanical interlocking [6,7]. In contrast to glass ceramics, different grinding procedures do not induce critical chipping in CAD/CAM RBCs. The high resistance of crack initiation may be due to a combination of lower hardness and lower elastic modulus [8]. While grinding increases the roughness of CAD/CAM restoration materials, it does also not impair the fracture resistance of these materials, even after aging [9]. SB the restoration material in order to improve adhesion, however, increases surface roughness while having the disadvantage of causing microcrack-related damage to CAD/CAM RBCs [10,11]. Yoshihara et al. showed that this damage was partly so severe, that additional silanization did not improve bond strength for the investigated material [10].

Regarding the luting materials for prosthetic restorations made of CAD/CAM RBC, the vast majority of studies in this area focus on manufacturers’ recommendations, resulting in predominant data for universal and self-adhesive resin cements [12]. Since no type of luting material category is suitable for the full broad spectrum of indirect restorative procedures, thorough understanding of each materials’ properties is necessary.

Zinc phosphate cement was introduced in 1879 and is one of the oldest luting cements known in dentistry [13]. It is a two-component acid—base reaction cement. The basic constituent of the powder (the zinc oxide) reacts with phosphoric acid in an aqueous environment. An exothermic reaction in addition to the initial low pH of the set cement can cause irritation of the tooth pulp [5,13]. The advantages of zinc phosphate cements include easy mixing and adequate strengths of the set cement for clinical use. However, these types of cements do not adhere to tooth and restorative material and are soluble in oral fluids [5,14,15].

Glass ionomer cements (GICs) are also acid-base reaction cements [13]. The three essential ingredients are polymeric water-soluble acid, basic glass, and water [16]. They show a number of advantages, such as fluoride release, chemical bonding to tooth structure, and bioactivity [16]. Still, conventional GICs are sensitive to both moisture and dehydration and have poor mechanical properties [17]. Adhesion to GICs can be attributed to self-etching through the polyacid component and also to ionic bonding between the carboxylate groups of the polyacid and the calcium ions of the tooth substrate [16].

In addition to the components of conventional GICs, resin-modified glass ionomer cements (RMGICs) also contain a monomer mixture with its associated initiator system [16]. The rapid radical polymerization reaction of the monomers protects the cement against premature exposure to water and dehydration [17]. The primarily setting reaction is therefore the radical polymerization followed by the slower acid-base reaction [16,18].

In contrast to conventional cements, adhesive resin cements (ARCs) have equally or superior mechanical properties, low solubility, and good esthetic qualities [19]. Depending on the adhesive system used to prepare the tooth before cementation, ARCs can be subdivided into three groups. This process can either be an etch-and-rinse adhesive, a self-etching primer, or even no conditioning at all if self-adhesive RCs are involved. The dominant setting reaction is the radical polymerization that can be initiated by light exposure or through a self-curing mechanism. In a recent study a self-adhesive and a conventional ARC showed comparable initial (24 h after setting) mechanical properties, but the conventional ARC performed better after aging [17].

Given the wide variety of luting materials available, the aim of this study was to evaluate the bonding strength and reliability of five luting material categories to a CAD/CAM RBC. In addition, the incidence of fractures and the index of cohesion with or without SB over time was considered.

We hypothesize that the following factors have no impact on shear bond strengths (SBS) and Weibull moduli: (1) pretreatment of the restoration material through SB (2) category of luting material used, and (3) in vitro aging.

## 2. Materials and Methods

This study examined the SBS of a CAD/CAM RBC (Grandio Blocs, Voco GmbH, Cuxhaven, Germany) with and without previous sandblasting (SB and NSB, respectively) to five different luting material categories: (1) temporary zinc phosphate cement, (2) GIC, (3) RMGIC, (4) self-adhesive RC, and (5) conventional ARC. Table 1 gives detailed information of all tested materials.

### 2.1. Specimen Preparation

To build the substrate for bonding, 25 CAD/CAM blocks (GB) were cut into 16 cuboids (7 mm × 7 mm × 4 mm) each, using a low-speed diamond saw (Isomet low-speed saw, Buehler, Leinfelden-Echterdingen, Germany) under water cooling. In total, 400 CAD/CAM RBC cuboids were embedded in methacrylate resin (Technovit 4004 transparent, Heraeus Kulzer GmbH, Wehrheim, Germany) using a stainless-steel cylinder as a mold. An additional 400 cylinders were obtained from the same CAD/CAM RBC (GD) through CAD/CAM milling (ø 3 × 18 mm) followed by cutting the obtained cylinders into smaller segments (ø 3 × 2 mm) using a low-speed diamond saw (Isomet low-speed saw). All CAD/CAM RBC surfaces underwent a finishing procedure with silicon carbide (SiC) 600 grit abrasive papers (Hermes, EXAKT Advanced Technologies GmBH, Norderstedt, Germany) under running water using a grinding and polishing machine (Leco Corp. SS-200, St. Joseph, MI, USA) [20]. Half of the embedded substrates (*n* = 200) and half of the cylinders (*n* = 200) were SB (50 µm aluminum oxide, duration: 10 s, angle: 90°, pressure 10^5^ Pa, distance: 5 mm) and subsequently cleaned for 5 min using an ultrasonic bath with distilled water. An adhesive system for preparing the CAD/CAM RBC was only necessary for BQM, which required a silane coupling agent (Ceramic Bond). Ceramic Bond was applied as a thin layer on both CAD/CAM RBC surfaces and allowed to take effect for 60 s after which time it was carefully dried with oil-free air. After the luting materials were mixed according to the manufacturer’s recommendations, they were applied on the cylinder with a micro-brush and subsequently placed on the embedded CAD/CAM RBC surface. Then a 4 kg static-load, that was measured with a scale (MAUL tronic S, Jakob Maul GmbH, Bad König, Germany), was applied by finger pressure for 5 s [21]. Excess material was carefully removed by a micro-brush. Before curing, BQM and BSE were covered with an oxygen-blocking gel (Liquid Strip, Ivoclar Vivadent, Schaan, Liechtenstein) to ensure polymerization of the outer resin layer. The materials BQM, BSE, and MQM were light-cured from the occlusal and two lateral sides for 20 s each with a blue-violet light-emitting diode (LED) curing unit (Bluephase^®^ Style, Ivoclar Vivadent; Irradiance = 1386 mW/cm², as assessed by a spectrophotometer, MARC System, Bluelight analytics, Halifax, NS, Canada). Figure 1 shows an illustration of a shear bond specimen.

The 400 resulting specimens were stored in a humid chamber for 1 h at 37 °C and then stored in distilled water at 37 °C for 23 h until SBS measurements to fulfill a 24 h storage interval. After 24 h of storage, half of the specimens from each group were thermally aged for 10,000 thermocycles in distilled water between 5 and 55 °C at a dwell time of 30 s per temperature and a transfer time of 10 s between baths (Willytec, Dental Research Division, Munich, Germany).

### 2.2. SBS Measurements

The specimens were fixed into a cylindrical base metal form and mounted with the bonding surface close to and parallel to the guillotine (thickness 2 mm) on the holding devices of a universal testing machine (Z 2.5, Zwick/Roell, Ulm, Germany). The test was conducted with a cross-head speed of 0.5 mm/min. SBS values were recorded for each specimen in units of MPa using the formula: SBS = F_max_/A in which F_max_ is the load at the moment of failure (N), and A is the bonding area of the specimen (7.0 ± 0.1 mm²), which was calculated for each individual specimen by measuring the diagonal of the bonded cylinder. Figure 2 illustrates the SBS measurement with the guillotine moving toward the cylinder.

### 2.3. Fractographic Analysis

The fracture surfaces were examined with a stereomicroscope (Stemi 508, Carl Zeiss AG, Oberkochen, Germany) in order to determine the failure mode. The surfaces were photographed using a microscope extension camera (Axiocam 305 color, Carl Zeiss AG) and were analyzed with computer software Image J (Java^®^, Sun Microsystems, Inc., Santa Clara, CA, USA). All specimens failed within the luting material-CAD/CAM RBC interface. These adhesive failures were further sub-classified: (1) adhesive, when the luting material de-bonded entirely from one surface (substrate or cylinder); (2) mixed, when the luting material de-bonded partly from both surfaces but the complementary luting material parts amount to 100%; and (3) cohesive, when failure occurred within the luting material and more than 50% of both surfaces appeared to be covered by cement remnants. Representative specimens were selected for scanning electron microscopy (SEM) analysis. Before the analysis the specimens had to be sputtered with a 58 nm thick gold-palladium layer (Polaron Range Sputter Coater SC7620, Quorum Technologies, Newhaven, England). Micrographs were taken using a SEM (Zeiss Supra 55 V P, Carl Zeiss AG) operating at 10 kV. Selected specimens were cut perpendicular to the feed direction of the guillotine with a low-speed diamond saw (Isomet low-speed saw), to allow direct sight of the substrate in depth. The cross-sectional area was then finished with SiC abrasive papers (1200 grit, 1500 grit, 2000 grit, 2500 grit and 4000 grit) (Hermes), after which it was polished with a diamond sprayed (DP-Spray, Struers GmbH, Puch, Austria) polishing cloth (DP-Pan 200 mm, Struers GmbH).

### 2.4. Statistical Analysis

The data were analyzed using IBM SPSS Statistics computer software (Version 25, International Business Machines Corporation, New York, NY, USA). A Kolmogorov–Smirnov test revealed a normal distribution of the data. Equality of variances was determined with Levene’s test (*p* = 0.356). One- and multiple-way analysis of variance (ANOVA) and Games-Howell honestly significant difference (HSD) post hoc tests (α = 0.05) were used. A multivariate analysis quantified the effect of the parameters material, aging, and surface pretreatment. Pearson’s chi-square test was used for the non-parametric parameters material, failure mode, aging, and pretreatment.

A Weibull analysis was performed to evaluate the reliability of bond strength values for the different materials. This model represents the cumulative probability of failure as follows [22]:$$F\left(\sigma \right)=1-\exp\left[-{\left(\frac{\sigma }{\sigma_{0}}\right)}^{m}\right]$$
where *F* is the probability of failure, *σ* represents the measured bond strength, *σ*_0_ the characteristic strength, at which the probability of failure is per definition 63% and m is the Weibull modulus. The double logarithm of this formula is:$$\ln\ln\frac{1}{1-F}=m\ln\sigma -m\ln\sigma$$

The logarithm of the measured bond strength (ln *σ*) is plotted against the double logarithm of the cumulative probability of failure (lnln [1/(1 − *F*)]) in a coordinate system. A linear regression line is calculated through the obtained coordinates. Its gradient represents the Weibull-modulus (m). The coefficient of determination (R²) describes additionally the goodness of fit of the regression. The standard error of the Weibull modulus (m) allows calculation of the 95% confidence interval, resulting from the formula 1.96 × standard error on both sides of the mean (m).

The sample size for this study was not calculated a priori. A post hoc power analysis was performed using a power analysis software (G*Power v3.1.9.7, Heinrich–Heine–Universität, Düsseldorf, Germany). With an effect size of 1.6 and an α error set at 0.05, the analysis revealed a power of 100%.

## 3. Results

### 3.1. Shear Bond Strength

The analysis showed a statistically significant (*p* = 0.005) interaction between aging, pretreatment, and luting material, in addition to two-way interactions between material and aging (*p* = 0.008), aging and pretreatment (*p* = 0.021), and between material and pretreatment (*p* < 0.001). The SBS results are illustrated in Figure 3. Table 2 summarizes the results of the one-way-ANOVA and Games-Howell test outcomes. It allows statistical comparison of the mean SBSs obtained depending on aging and pretreatment used (uppercase letters refer to within row comparisons), as well as luting material used (lowercase letters refer to within column comparisons). Identical letters identify groups in rows and columns that are statistically similar.

The results have identified higher SBS for the SB groups compared to the NSB groups, both after 24 h and aging, in all materials—except PQM. The difference between the pretreatment groups was high for BQM (*p* < 0.001), MQM (*p* < 0.001), and M (*p* < 0.001). The significance slightly decreased for BSE from *p* < 0.001 in the 24 h group to *p* = 0.012 in the aged group, which indicates a greater decrease in the bond strength after aging in the SB group.

Due to the high amount of pre-test failures after aging for M (10% (SB) and 75% (NSB)) and PQM (100%), samples which failed during aging were assigned the value F_max_ = 0 N. Samples that still bonded after aging but failed before testing (such as during mounting into the test device) were assigned the value F_max_ = 0.01 N, as the lowest value that could have been measured for these materials lies above (F_max_ = 0.018 N). A significant decrease in SBS after aging was identified for M (NSB: *p* < 0.001; SB: *p* = 0.041) and PQM (NSB and SB: *p* < 0.001) in addition to the SB-group of BQM (*p* = 0.042).

When comparing the materials, BQM and BSE performed in a statistically similar manner in all pretreatment and aging groups and had by far the strongest bond strengths. The results of MQM were in between the cements and the composite luting materials. The analysis identified the weakest SBS when using PQM. M shows significantly lower bond strengths than MQM but higher bond strengths than PQM, except for the NSB aged group.

### 3.2. Weibull Analysis

The Weibull distribution is illustrated in Figure 4. Corresponding Weibull moduli (m) and coefficients of determination (R²) are shown in Table 3.

The reliability was strongly influenced by the surface pretreatment, except for the aged BSE group for which the same reliability for NSB and SB was found. All materials lost reliability with aging, while the NSB group of BQM was the only group that showed a slight but significant increase in reliability after aging. Both ARCs (BQM and BSE) and MQM were statistically similar after 24 h, but BQM showed the highest reliability after aging. The lowest Weibull moduli were noted for the conventional cements M and PQM, especially after aging.

### 3.3. Fractographic Analysis

Evaluation of surface morphology under light microscope identified that all failures occurred adhesively. Consequently, all results were valid for SBS calculations. Obvious subsurface damage (visible cracking running into the CAD/CAM RBC) was observed after the SBS-test only in the SB of the BQM and BSE groups in the majority of cases (Table 4). A one-way ANOVA showed an interactive effect between SBS and subsurface defects (*p* < 0.001). SB ARC samples without this feature showed no correlation between SBS and failure mode (*p* = 0.093). A non-parametric Pearson’s chi -square test revealed a correlation of material and failure mode (*p* < 0.001) as well as pretreatment and failure mode (*p* < 0.001) but no correlation with failure mode and aging (*p* = 0.473).

Figure 5 shows the predominant fracture modes, adhesive and mixed. The cohesive failure mode, with failures that occurred within the luting material, was observed only for PQM (Table 4). No cohesive failures within the substrate (CAD/CAM RBC) were found. The subsurface defects could be seen more clearly under a light microscope (Figure 6A) than on SEM images (Figure 6B) due to crack shadowing. The round area, where the CAD/CAM RBC cylinder was bonded to, is surrounded by luting material remnants (Figure 6A,B). Figure 6A shows the subsurface damage in the lower half of the bonded area (black arrows). For evaluation of defect depth, one sample with this feature was cut vertically (cross-section) at the damaged location (Figure 6C,D). Evidently, not only the superficial area was damaged, but the CAD/CAM RBC was also damaged in depth. Figure 6C shows both the roughened surface of the CAD/CAM RBC through sandblasting (black arrow) and the damaged subsurface in depth (white arrow). Figure 6D allows direct view on the cracks depth (90° rotation to Figure 6A,B). The white horizontal area in the middle of the figure suggests damage to the CAD/CAM RBC up to approximately 40 µm from the surface. The darker region below depicts undamaged CAD/CAM RBC; the undefined area above represents remnants of luting material in the background.

## 4. Discussion

The purpose of this in vitro study was to evaluate the influence of SB on the bonding reliability of five different luting materials to a CAD/CAM RBC. As SB improves bond strengths, the first null hypothesis can be rejected for all tested luting materials, except the zinc phosphate cement. This finding indicates that SB had a positive effect on the bond strength between the CAD/CAM RBC and the luting material, which has been confirmed in recent studies [1,2,6,12].

Subsurface defects in the CAD/CAM RBC were found to be exclusively in the SB specimens of the ARCs (Table 4). Although the CAD/CAM surface remained intact, their presence can, however, be interpreted as a preliminary stage toward a cohesive fracture that may occur in the CAD/CAM RBC. It has been previously proven that cohesive failures in the substrate may be related to the mechanics of the test method and are amplified by contact friction [23]. Since the location of these subsurface cracks is always the same (the lower half of the bonded area), one has to conclude that they are also due in part to the mechanics of the SBS test. The failure path is also unrestricted and leads into the substrate, in contrast to the debonding along an interface at the adhesive failures [23]. The inhomogeneous surface of the SB samples led to a higher contact friction, i.e., higher bond strengths, and as a result to a higher tendency for the failure path to grow into the CAD/CAM RBC. SB has also been proven to cause damages to the substrate [10,11], which can also lead to a higher tendency for subsurface cracks to occur at the corresponding weak points and cause deepening of any existing microcracks. This observation suggests that the presence of these subsurface cracks is evidence that microcracks can be found all over the SB surface of the CAD/CAM RBC, which due to the mechanics and the distribution of forces during the SBS test, will lead to defects at the corresponding weak points. These microcracks may have major clinical relevance by diminishing the long-term fatigue behavior of the material, which would lead to catastrophic failure [10]. An indication for this process was the smaller difference in SBS values between SB and control groups after aging. In view of the above findings, one should also question the selected formula used to calculate bond strength, which is inaccurate for SB samples since the bonded area (A) drastically increased after SB, and the increased proportion of stress peaks falsified a homogeneous calculation of the formula, σ = F_max_/A.

The second null hypothesis needs to be rejected, since the different categories of luting materials performed unequally. The temporary zinc phosphate cement, PQM, showed a weak performance with respect to SBS. Since zinc phosphate cements lack adhesion, they might clamp the restoration to the tooth through undercuts and retention [5,14]. In the present experimental design, however, no possibility for retention was found. As SB did not induce an improvement in SBS values, micromechanical interlocking does not seem to be possible for this luting material. Failures of PQM occurred within the cement (cohesive) rather than at the interface. As a result, the bonding values obtained in this experiment are in fact not measures of adhesive bond strength but of the tensile strength of the cement [16].

The GIC Meron performed better regarding SBS especially after SB and aging. Studies refer to the improved retention of GICs as a result of their adhesiveness to polar substrates of the restoration material [15]. Mujdeci et al. showed that bond strength values of a conventional GIC to enamel and dentin increased with airborne particle abrasion of the substrate. The mean SBS in the above-mentioned study varied from 5.5 MPa in the control group to 7.1 MPa in the SB group, which is similar to our results [24].

With respect to the RMGIC cement, MQM, the additional use of monomers in a traditional GIC seem to clearly improve the bonding to CAD/CAM RBC. Accordingly, the SBS values obtained when four RMGIC cements were used as luting materials were higher than those achieved with conventional GIC and lower than those obtained when ARC were used [25]. It has been shown that the bonding mechanism is based on the formation of an ionic carboxylate bond between the carboxyl groups of the methacrylated polyalkenoic acid of the paste-liquid RMGICs with the polar substrates of the restoration material [18].

The incorporation of acid-functionalized methacrylate or related monomers is a critical component in self-adhesive RCs because effective chemical bonding requires a polyacid matrix structure [13,26]. With respect to substrates other than teeth, only a few studies comparing the bond strength of self-adhesive RCs to ceramics and CAD/CAM RBCs have been published. In a study by Ferracane et al., the bond strength of a self-adhesive RC to zirconia was shown to be equivalent to that of the conventional ARC, and the bonds of both were enhanced by pretreating the ceramic with SB [26]. This finding is similar to the present results, showing higher SBS values after SB and no statistical difference of BSE to BQM. In case of BQM, the silan coupling agent Ceramic Bond interacts with the silanol groups (Si-OH) of the fillers on the composite surface. It has been also proved that silane coupling agents can form linkages to the alumina particles deposited on the substrate surface after sandblasting (Al-O-Si) [27]. However, Al-O-Si-bonds were shown to be susceptible to hydrolysis [28], which might be a possible reason for the significant decrease of SBS after aging. Yano et al. proved that airborne-particle abrasion causes chemical changes on the CAD/CAM RBC surface by reducing the density of the silanol groups. As a consequence, SB leads to an improvement in bond strengths due to the increment on surface roughness and combined use of sandblasting and silanization have no synergistic effect [7]. On the other hand, Nagasawa et al. found higher bond strengths for most analyzed CAD/CAM RBCs to a self-adhesive RCs when SB was combined with silanization, which is contrary to the present results [29].

A long-term adhesive bond under clinical conditions can only be achieved if the luting materials used are resistant to water sorption and changes due to temperature fluctuations [15,19]. Repetitive expansion and contraction stresses between dissimilar materials in addition to hydrolytic-degradation therefore could be a reason for the statistically relevant decrease in SBS after aging for the conventional cements. It does not seem to be efficient to use conventional cements with CAD/CAM RBC to achieve durable bond strengths, regardless of the pre-treatment procedure. However, for the RMGIC, MQM, no changes in SBS values after aging were detected. RMGICs are less brittle than GIC and show improved mechanical properties over time due to further maturation and mechanical stability through the polymer [17]. The BQM and BSE ARCs showed a decrease in SBS values after aging exclusively in the SB groups. 10,000 thermocycles have been estimated to represent one service year in vivo but are certainly an arbitrary number [30]. Taking into consideration the lifespan of a definitive restoration, an even more significant decrease might occur if this number were to be increased. Therefore, the third null hypothesis can be rejected.

The analytical approach to pre-test failures in the literature is inconsistent. A review analyzing 147 references identified that the majority of papers have included pre-test failures as zero values in the statistical analysis [20]. In the present study, samples that failed during aging were also included in the statistics and assigned the value 0 MPa. Some samples survived the thermal aging process, which in fact is proof of higher bond strengths, but failed while mounting into the test device. These samples were assigned the value F_max_ = 0.01 N, as the lowest value that could have been measured lies above (F_max_ = 0.018 N). Similar to this approach, Inokoshi et al. reported the number of failures during mechanical aging and included them in the statistics. Since these failures did have some bond strength, a random value between 10 MPa and the lowest value measured in the respective group was assigned [31].

All samples in this study were stored and aged in distilled water. This decision was made due to the fact that distilled water is the medium described in all standards for testing the bond strength and mechanical properties of dental materials. Since significantly more ions were leached from resin-based composite fillers in artificial saliva than in distilled water [32], greater decrease in adhesive strength would be expected with aging in artificial saliva. However, a recent study investigating the influence of aging in artificial saliva compared to distilled water on the bond strength of resin-based composites demonstrated the opposite [33]. Furthermore, the decision for the choice of medium also concerns the fact that the mechanical properties of resin-based composites [34] and glass ionomer cements [35] were not affected by the storage medium, which was either distilled water or artificial saliva.

SB increased the reliability of all luting materials after 24 h, while aging led to a decrease in Weibull moduli for most groups. For both ARCs, the decrease of Weibull moduli after aging was higher in the SB groups compared to the NSB groups. BQM showed the highest reliability after aging (NSB: m = 5.2 ± 0.7; SB: m = 4.1 ± 0.4). The lowest Weibull moduli were recorded for the conventional cements M and PQM. This may be due to the more frequent presence of pores and flaws in these conventional cements compared to ARCs [17]. When these pores randomly accumulate in the tensile zone of the bonding interface, a faster shearing of the cylinder may be probable.

A comparison between light microscopy and SEM showed that the subsurface defects were distinctly easier to detect under the light microscope due to the shadow formation while using a vicinal illumination (Figure 6A,B). Therefore, the optical examination is a crucial step for setting the fractographic foundation for SEM analysis while SEM has its own advantages in discovering details, such as fine hackle lines and material flaws [36].

## 5. Conclusions

Within the limitations of the present experiment, several conclusions can be drawn:SB the CAD/CAM RBC leads to higher bond strength values to ARCs, resin-modified, and conventional GICs.SB leads to microcracks in the CAD/CAM RBC, which can deepen and lead to catastrophic failure at certain shear forces.The conventional ARC BQM showed the highest reliability for bonding to the analyzed CAD/CAM RBC.Cracks after SBS testing are easier to detect under a light microscope than on SEM images due to shadow formation.

Further studies are necessary to evaluate if the in vivo performance of CAD/CAM RBC restorations is compromised through SB.

## Figures and Tables

**Figure 1 jfb-13-00217-f001:**
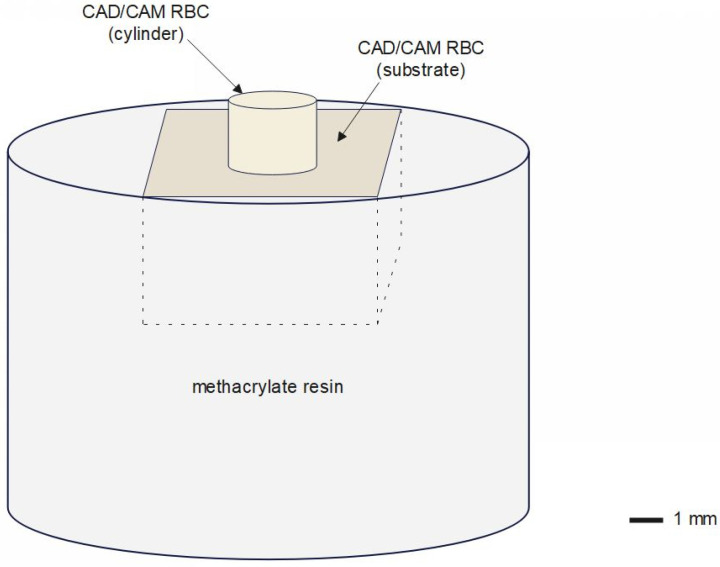
Illustration of a shear bond specimen.

**Figure 2 jfb-13-00217-f002:**
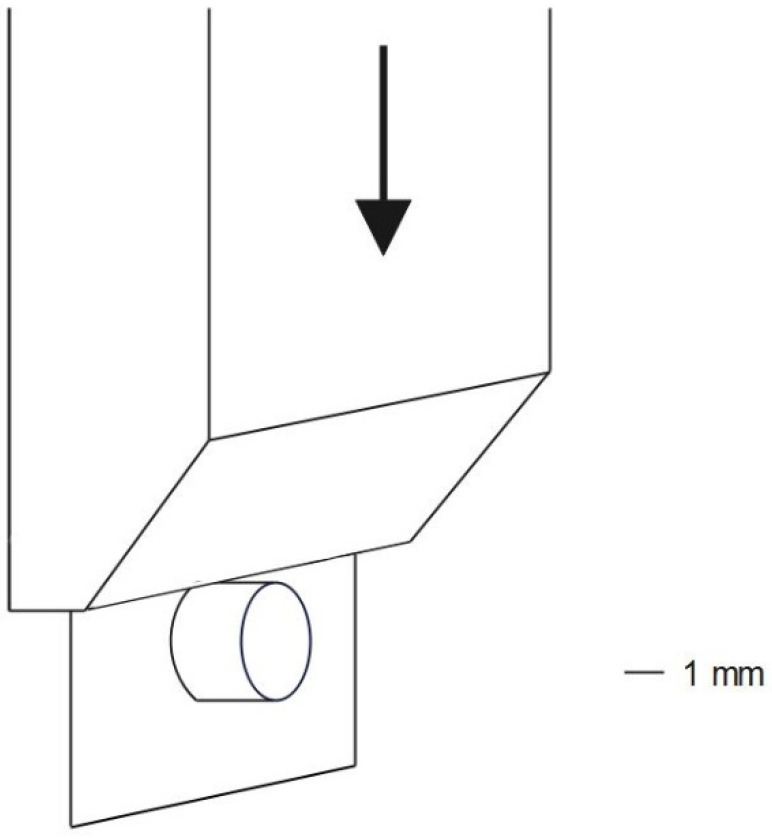
Illustration of SBS measurement.

**Figure 3 jfb-13-00217-f003:**
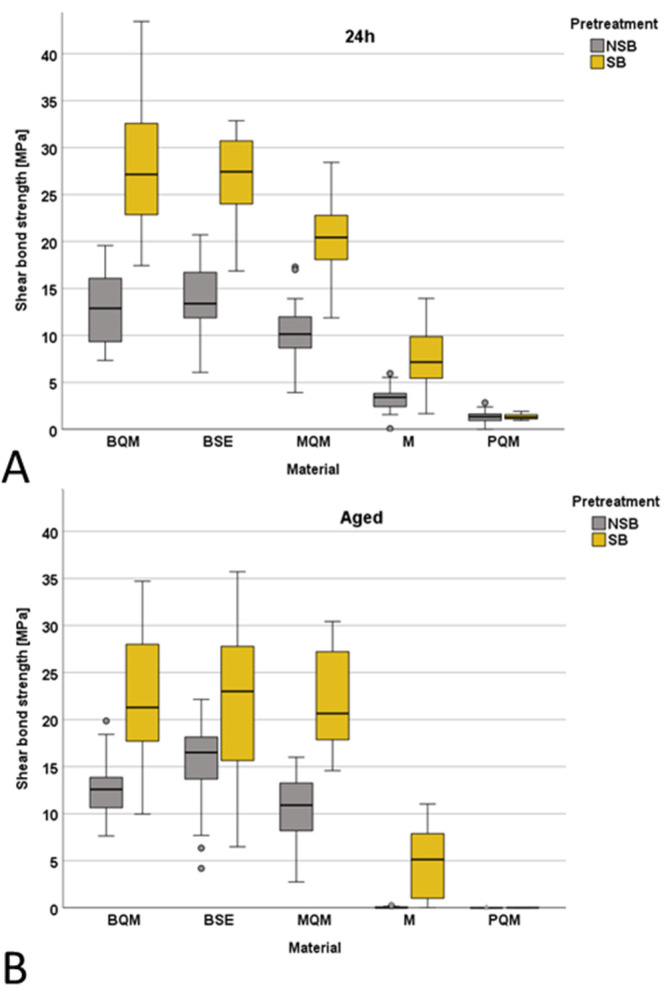
(**A**) Box plot mean shear bond strength (SBS) after 24 h; (**B**) Box plot mean shear bond strength after aging.

**Figure 4 jfb-13-00217-f004:**
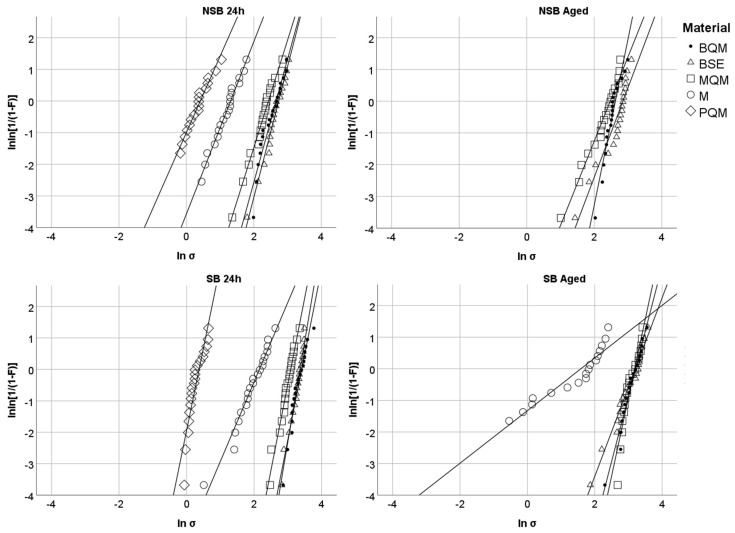
Weibull distribution depending on pretreatment (NSB/SB) and aging (24 h/aged).

**Figure 5 jfb-13-00217-f005:**
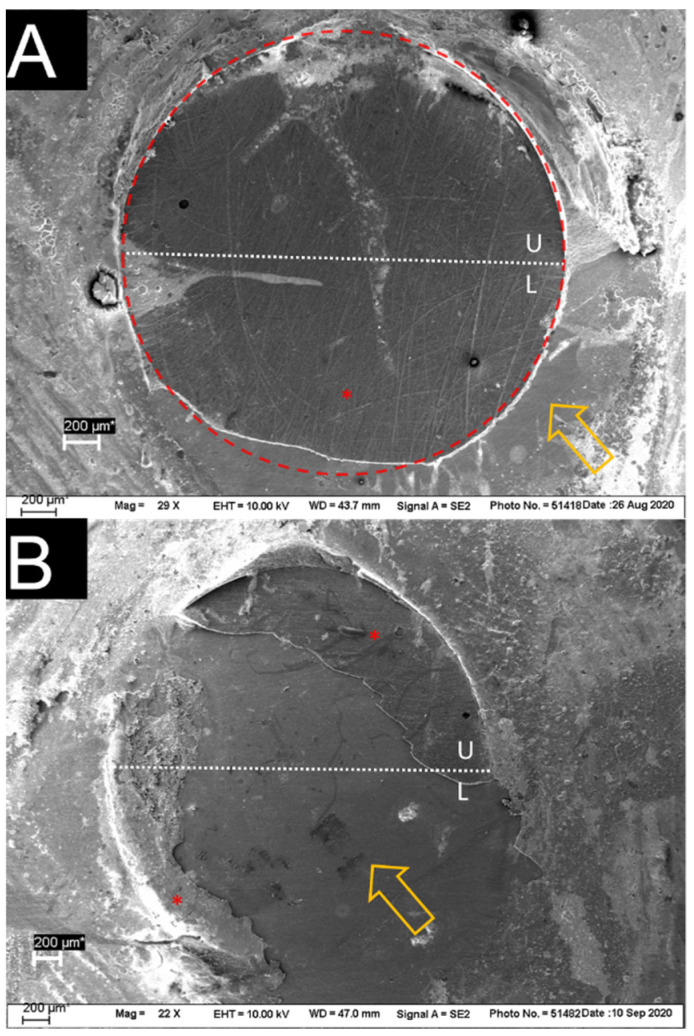
(**A**) adhesive fracture of BQM, NSB, 24 h; (**B**) mixed fracture of BSE, NSB, 24 h; U = upper half, L = lower half of the bonded area; “*”: luting material.

**Figure 6 jfb-13-00217-f006:**
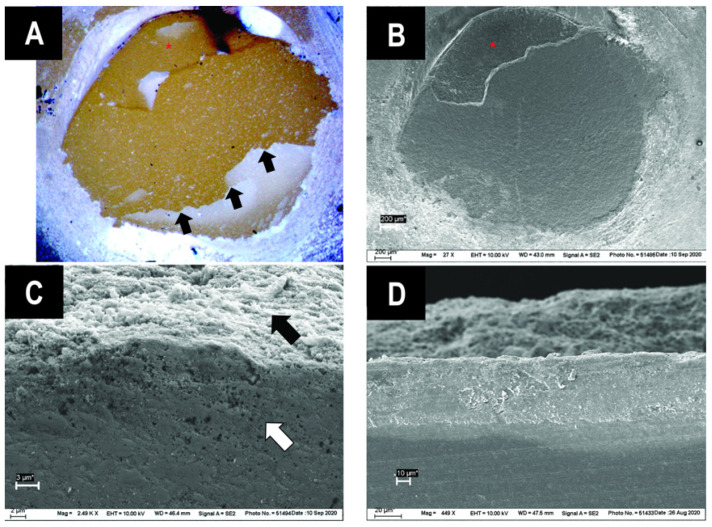
Sandblasted samples with subsurface damage after SBS test. (**A**) Light microscope image of BSE, showing a clear crack of the CAD/CAM RBC (black arrows); (**B**) SEM image of the same sample; (**C**) cross-section of BSE, black arrow identifies sandblasted surface and white arrow shows subsurface damage; (**D**) cross-section of a crack identifies the damaged CAD/CAM material and the depth of the crack.

**Table 1 jfb-13-00217-t001:** Analyzed materials: code, material, material type, lot number, and composition according to the manufacturer. All materials were manufactured by the company VOCO GmbH (Cuxhaven, Germany).

Code	Material	Type	Lot №	Composition
GB	Grandio Blocs	CAD/CAM RBC	1905635	86% anorganic filler in a polymer matrix (14% UDMA + DMA)
GD	Grandio Disc	CAD/CAM RBC	1838335	86% anorganic filler in a polymer matrix (14% UDMA + DMA)
PQM	Provicol QM Plus	Temporary zinc phosphate cement	1907633	catalyst: zinc oxide, Ca(OH)₂, MgO, oil; base: fatty acids, modified resin, oil, SiO_2_
M	Meron	Glass ionomer cement	1906388	reactive glass, pigments, polyacrylic acid, tartaric acid, H_2_O
MQM	Meron Plus QM	Resin-modified glass ionomer cement	1908135	methacrylates, BHT, peroxide, reactive glass, SiO_2_, pigments, polyacrylic acid, tartaric acid, H_2_O
BSE	Bifix SE	Dual-curing self-adhesive resin cement	1910348	catalyst: dimethacrylates, phosphoric acid, dimethacrylate ester, methacrylates, BPO, SiO_2_, BAS glass ceramic, BHT; base: dimethacrylates, methacrylates, CQ, amine, SiO_2_, BAS glass ceramic, BHT; filler content: 66.3%
BQM	Bifix QM	Dual-curing conventional adhesive resin cement	1905094	catalyst: dimethacrylates, BPO, SiO_2_, BAS glass ceramic, BHT; base: dimethacrylates, CQ, amine, SiO_2_, BAS glass ceramic, BHT; filler content: 70%
-	Ceramic Bond	Silanization agent	1843482	organic acids, 3-methacryloxypropyl-trimethoxysilane, acetone

UDMA: urethane dimethacrylate, DMA: dimethylacetamide, Ca(OH)₂: calcium hydroxide, MgO: magnesium oxide, SiO_2_: silicon dioxide, H_2_O: water, BHT: butylated hydroxytoluene, BPO: benzoyl peroxide, BAS: barium aluminum silicate, CQ: camphorquinone.

**Table 2 jfb-13-00217-t002:** Shear bond strength (SBS) in units of MPa (mean ± standard deviation) as a function of luting material, pretreatment, and aging methods (One-way analysis of variance [ANOVA] and Games–Howell, at α = 0.05). Uppercase letters refer to within row comparisons, while lowercase letters refer to within column comparisons. Identical letters identify groups that are not statistically significantly different. For each group *n* = 20.

Aging	24 h		Aged	
Material/ Pretreatment	NSB	SB	NSB	SB
BQM	13.0 ± 3.9 ^A ab^	28.0 ± 6.3 ^B a^	12.7 ± 3.0 ^A ab^	22.5 ± 6.2 ^C a^
BSE	14.1 ± 3.9 ^A a^	26.8 ± 4.7 ^C a^	15.1 ± 4.6 ^AB a^	21.9 ± 7.8 ^C a^
MQM	10.2 ± 3.4 ^A b^	20.2 ± 4.2 ^B b^	10.5 ± 3.8 ^A b^	22.1 ± 5.2 ^B a^
M	3.3 ± 1.4 ^A c^	7.6 ± 3.0 ^B c^	<0.1 ± 0.1 ^C c^	4.6 ± 3.7 ^A b^
PQM	1.3 ± 0.7 ^A d^	1.3 ± 0.3 ^A d^	<0.1 ± <0.1 ^B c^	<0.1 ± <0.1 ^C c^

**Table 3 jfb-13-00217-t003:** Weibull modulus with standard error (Std. Error) and R square (R²) values for each mate rial, aging, and surface pretreatment.

Aging	24 h			Aged	
Material/Pretreatment	NSB		SB	NSB	SB
BQM	m	3.9 (0.26)	5.4 (0.33)	5.2 (0.36)	4.1 (0.18)
	R²	0.93	0.94	0.92	0.97
BSE	m	4.1 (0.15)	6.4 (0.24)	2.8 (0.22)	2.8 (0.12)
	R²	0.98	0.98	0.90	0.97
MQM	m	3.4 (0.12)	5.5 (0.22)	2.7 (0.10)	5.0 (0.40)
	R²	0.98	0.97	0.98	0.89
M	m	1.2 (0.16)	2.5 (0.11)	-	0.6 (0.05)
	R²	0.74	0.97	-	0.93
PQM	m	2.4 (0.11)	5.3 (0.51)	-	-
	R²	0.71	0.86	-	-

**Table 4 jfb-13-00217-t004:** Distribution of various failure types and percentage of subsurface defects for not-sandblasted (NSB) and sandblasted (SB) samples (%).

Pretreatment	Material	Adhesive	Mixed	Cohesive	Subsurface Defects
NSB	BQM	62.5	37.5	0	0
	BSE	52.5	47.5	0	0
	MQM	40	60	0	0
	M	92.5	7.5	0	0
	PQM	0	2.5	97.5	0
SB	BQM	25	75	0	65
	BSE	20	80	0	67.5
	MQM	32.5	67.5	0	0
	M	62.5	27.5	0	0
	PQM	0	10	90	0

## Data Availability

Not applicable.

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
