# Peer review of "Adhesion to a CAD/CAM Composite: Causal Factors for a Reliable Long-Term Bond"

_jfb, 2022, doi:10.3390/jfb13040217_

Round 1

Reviewer 1 Report

Dear authors,

Please, see the comments below on points that I consider important for making your work improved.

The introduction does not explain the significance of the analyzed parameters (SBS, sandblasting and Weibull modulus) for the materials. 

In section 2, some relevant information is missing. 

Line 78: the abbreviation (SB) should be explained

Line 85: the abbreviation (SBS) should be explained

Line 91: no space

In Tab.1: LOT-  make sure uppercase letters are required, standardization in the description of the table is suggested

Line 95 and 99: it is recommended to standardize the sample size units, e.g. mm

Line 105: it is recommended to standardize the time unit, s

Line 106 and 113: it is recommended the use of SI units

Line 107: it is recommended to state the name of the producer of distilled water

Lines 126 to 130 describe the aging procedure, not SBS, it is recommended to separate the text.

how was the humidity measured ?, device ?, accuracy ?,

how accurately was the temperature measured?

Line 132: Please enter the dimensions of the guillotine. 

Line 134: Why a cross-head speed was 0.5 mm / min?

Line 136: Why is the area given as 7.0 (accuracy?) The surface area of the samples after cutting and sandblasting was not measured. Were the samples measured before the test, after aging process? 

In the section: SBS measurements, it is recommended to include a diagram or a photo of the test.

In section 2.4: The computational assumptions for the analysis of the Weibull module should be specified.

In Table 2. Please explain the signs A, a B, b, etc.

Line 196 to 202: The proposed solution is questionable. If the samples are damaged before the shear test or during assembly, they should not be included in the results. The given value of 0.01 N for a machine with a measuring range of 2500 N is a value of error of machine.

In Table 3, it is suggested to delete the value 0.

Fig. 3. Please explain the notations F and σ.

Line 296: The formula is theoretical, but a proper physical analysis of the dimensions of the samples was not performed prior to the test. Pure shear was assumed, but the tension and compression zones are marked in Figure 4, which suggests a bending effect. Please explain or correct it.

In section discussion: Results were not referenced to oral conditions The meanings of 10,000 thermocycles, the use of distilled water and temperature ranges are not given. No information is given on what shear values are generated under oral conditions. What shear strength values are required for dental materials? Basically, only the obtained results were compared, there is no description of their usefulness.

Author Response

The authors would like to thank the reviewers for taking the time to read and critically appraise the manuscript and for their positive, constructive comments on improving the work.

All comments to the corresponding author have been addressed independently below. The authors’ rebuttal is BLUE, and changes to the revised manuscript in light of the reviewers’ comments are presented in RED.

Comments and Suggestions for Authors

Dear authors,

Please, see the comments below on points that I consider important for making your work improved.

The introduction does not explain the significance of the analyzed parameters (SBS, sandblasting and Weibull modulus) for the materials. 

Author’s response: we extended the explanation for the measured parameters – please consider the changes in the manuscript (detail description of the Weibull statistics and the meaning of all terms in material and methods and changes during the manuscript).

In section 2, some relevant information is missing. 

Line 78: the abbreviation (SB) should be explained

Author’s response: SB was explained before in line 39: As a result, roughening of the surface by airborne-particle abrasion, hereafter referred to as sandblasting (SB), has been established to enable micromechanical interlocking.

Line 85: the abbreviation (SBS) should be explained

Author’s response: SBS was explained before in line 81-82: We hypothesize that the following factors have no impact on shear bond strengths (SBS) […].

Line 91: no space

Author’s response: Thank you for this observation. We corrected it accordingly.

In Tab.1: LOT-  make sure uppercase letters are required, standardization in the description of the table is suggested

Author’s response: LOT in the table was changed to Lot â„–.

Line 95 and 99: it is recommended to standardize the sample size units, e.g. mm

Author’s response: we corrected accordingly.

Line 105: it is recommended to standardize the time unit, s

Author’s response: Corrected as suggested.

Line 106 and 113: it is recommended the use of SI units

Author’s response: Corrected as suggested; kgf was changed in kg and 1 bar in 105 Pa.

Line 107: it is recommended to state the name of the producer of distilled water

Author’s response: Distilled water is considered a type of purified water. Its way of production ensures the same quality and type of water after boiling since impurities are eliminated. It is produced by the clinical hospital and used in all laboratories. Thus, the name of the producer of distilled water is never mentioned in scientific work. Moreover, as we did not perform an analytical method it is of low importance.  

Lines 126 to 130 describe the aging procedure, not SBS, it is recommended to separate the text.

Author’s response: The respective lines were corrected, thank you for pointing this out.

how was the humidity measured ?, device ?, accuracy ?, how accurately was the temperature measured?

Author’s response: The respective sentence was corrected.

Line 132: Please enter the dimensions of the guillotine. 

Author’s response: The dimensions of the guillotine were added to materials and methods.

Line 134: Why a cross-head speed was 0.5 mm / min?

Author’s response: A literature review by Scherrer et al.* compared the bond strength measurements and results obtained by 147 different studies. They criticized the high scatter in dentin bond strengths regardless of the adhesive and the bond test used (shear, micro shear, tensile, and micro tensile). This may be due in part to the different study designs, which showed varying crosshead speeds, surface finishings and times prior to bond strength testing. Most reviewed studies that used a shear test for bond strength measurements chose a cross-head speed of 0.5 mm/min. We chose the most common cross-head speed (0.5 mm/min) to make our results more comparable to similar studies and reduce the scatter in data.

* Scherrer SS, Cesar PF, Swain MV. Direct comparison of the bond strength results of the different test methods: A critical literature review. Dent Mater 2010; 26(2):e78-e93. https://doi.org/10.1016/j.dental.2009.12.002.

Line 136: Why is the area given as 7.0 (accuracy?) The surface area of the samples after cutting and sandblasting was not measured. Were the samples measured before the test, after aging process?

Author’s response: The bonding area was defined by the CAD/CAM RBC cylinder bonded to the substrate since excess materials were carefully removed before the luting materials hardened. The diameter of the cylinders was 3 mm (± 0.01 mm), which results in a bonding area of 7 mm calculated with the formula A = π·r2. The diameter was measured for each cylinder individually.

In the section: SBS measurements, it is recommended to include a diagram or a photo of the test.

Author’s response: An Illustration of the SBS measurements was added.

In section 2.4: The computational assumptions for the analysis of the Weibull module should be specified.

Author’s response: The following paragraph was added:

Weibull analysis was performed to evaluate the reliability of bond strength values for the different materials. This model represents the cumulative probability of failure as follows […].

The logarithm of the measured bond strength (ln σ) is plotted against the double logarithm of the cumulative probability of failure (lnln [1/(1-F)]) in a coordinate system. A linear regression line is calculated through the obtained coordinates. Its gradient represents the Weibull-modulus (m). Additionally, the coefficient of determination (R²) describes the goodness of fit of the regression.

In Table 2. Please explain the signs A, a B, b, etc.

Author’s response: The letters are explained in the description of the table: Uppercase letters refer to within-row comparisons, while lowercase letters refer to within-column comparisons. Identical letters identify groups that are not statistically significantly different.

Line 196 to 202: The proposed solution is questionable. If the samples are damaged before the shear test or during assembly, they should not be included in the results. The given value of 0.01 N for a machine with a measuring range of 2500 N is a value of error of machine.

Author’s response: We strongly disagree with not including failed samples in the statistics. Please consider that we intended to highlight that some of the specimens survived the thermal aging process and some did not, as explained in the manuscript, which in fact is proof that a bond existed. Nonetheless, the value of 0.01 N is too low to have an impact on the mean shear bond strengths or the statistics of the respective luting materials. In conclusion, if these specimens were to be excluded or given the value 0 N, the results and statistics would have been the same. However, we adapted the Weibull plot accordingly and eliminated the 0.01 N bond-strength Force data and the corresponding strength for more clarity.

In Table 3, it is suggested to delete the value 0.

Author’s response: The respective values were deleted.

Fig. 3. Please explain the notations F and σ.

Author’s response: Explanation was added in chapter 2.4 Statistical analysis.

Line 296: The formula is theoretical, but a proper physical analysis of the dimensions of the samples was not performed prior to the test. Pure shear was assumed, but the tension and compression zones are marked in Figure 4, which suggests a bending effect. Please explain or correct it.

Author’s response: The respective figure was corrected. Since we assume pure shear, tension and compression zones were changed into the upper and lower half of the bonding area.  

In section discussion: Results were not referenced to oral conditions The meanings of 10,000 thermocycles, the use of distilled water and temperature ranges are not given. No information is given on what shear values are generated under oral conditions. What shear strength values are required for dental materials? Basically, only the obtained results were compared, there is no description of their usefulness.

Author’s response: The meaning of 10,000 thermocycles was added.

Reviewer 2 Report

A well-written manuscript with a sound study design and appropriate statistics. The argument discussed in this research has a direct clinical implication. I only suggest a minor revision about the acronyms used to label the materials: "RBC" is used for both CAD/CAM composite blocks and luting cement of group 4 and 5; "ARC" could be more indicated for adhesive resin cements.

Author Response

The authors would like to thank the reviewers for taking the time to read and critically appraise the manuscript and for their positive, constructive comments on improving the work.

All comments to the corresponding author have been addressed independently below. The authors’ rebuttal is BLUE, and changes to the revised manuscript in light of the reviewers’ comments are presented in RED.

Comments and Suggestions for Authors

A well-written manuscript with a sound study design and appropriate statistics. The argument discussed in this research has a direct clinical implication. I only suggest a minor revision about the acronyms used to label the materials: "RBC" is used for both CAD/CAM composite blocks and luting cement of group 4 and 5; "ARC" could be more indicated for adhesive resin cements.

Author’s response: Thank you for the appreciation. We followed your suggestion and changed RBC luting materials into adhesive resin cements (ARC).

Reviewer 3 Report

Dear Authors, this paper about the causal factors for a reliable long term bond to CAD/CAM composites is really interesting and could really be helpful to clinicians and to the scientific community.

Some issues need to be solved before its final publication:

Abstract: please divide abstract into small chapters: introduction, materials and methods, results, conclusion

Introduction: Overall is well written, please add a small chapter about how grinding and polishing CAD/CAM materials in order to improve adhesion could affect the fracture resistance, this is really important to help the readers better understand the behavior of these materials after being treated   as clinicians would do. This reference can help you: Ludovichetti FS, Trindade FZ, Adabo GL, Pezzato L, Fonseca RG. Effect of grinding and polishing on the roughness and fracture resistance of cemented CAD-CAM monolithic materials submitted to mechanical aging. J Prosthet Dent. 2019 May;121(5):866.e1-866.e8.

Materials and methods and results part is well written and easy to understand

Discussion: This part is fine, please check some minor orthographic english errors

Conclusion: Please add a final small sentence stating if further studies are needed or if this study is in agreement with literature.

Author Response

The authors would like to thank the reviewers for taking the time to read and critically appraise the manuscript and for their positive, constructive comments on improving the work.

All comments to the corresponding author have been addressed independently below. The authors’ rebuttal is BLUE, and changes to the revised manuscript in light of the reviewers’ comments are presented in RED.

Comments and Suggestions for Authors

Dear Authors, this paper about the causal factors for a reliable long term bond to CAD/CAM composites is really interesting and could really be helpful to clinicians and to the scientific community.

Some issues need to be solved before its final publication:

Abstract: please divide abstract into small chapters: introduction, materials and methods, results, conclusion

Author’s response: The suggested division of the abstract into small chapters is unfortunately in contrast to the „Instructions for Authors“: The abstract should be a single paragraph and should follow the style of structured abstracts but without headings.

Introduction: Overall is well written, please add a small chapter about how grinding and polishing CAD/CAM materials in order to improve adhesion could affect the fracture resistance, this is really important to help the readers better understand the behavior of these materials after being treated   as clinicians would do. This reference can help you: Ludovichetti FS, Trindade FZ, Adabo GL, Pezzato L, Fonseca RG. Effect of grinding and polishing on the roughness and fracture resistance of cemented CAD-CAM monolithic materials submitted to mechanical aging. J Prosthet Dent. 2019 May;121(5):866.e1-866.e8.

Author’s response: This part was added: […] In contrast to glass ceramics, different grinding procedures do not induce critical chipping in CAD/CAM composites. The high resistance of crack initiation may be due to a combination of lower hardness and lower elastic modulus […]. While grinding increases the roughness of CAD/CAM restoration materials, it does not impair the fracture resistance of these materials, even after aging [..]. SB the restoration material in order to improve adhesion, however, increases surface roughness while having the disadvantage of causing microcrack-related damage to the CAD/CAM RBCs […]. Yoshihara et al. showed that this damage was partly so severe, that additional silanization did not improve bond strength for the respective material […].

Materials and methods and results part is well written and easy to understand

Author’s response: Thank you for the appreciation.

Discussion: This part is fine, please check some minor orthographic english errors

Author’s response: Thank you for the appreciation. We checked the manuscript accordingly.

Conclusion: Please add a final small sentence stating if further studies are needed or if this study is in agreement with literature.

Author’s response: The following statement was added: Further studies are necessary to evaluate if the in-vivo performance of the analyzed CAD/CAM RBC  is compromised through SB.

Reviewer 4 Report

This manuscript could be an interesting study evaluating the bonding efficiency of various types of luting cements to CAD/CAM resin-based composites. However, the study suffers from an inappropriate selection of materials and numerous shortcomings and errors that made me recommend its rejection from such a highly rated scientific journal. I will try to show some of them.

1.                  Choice of luting materials - e.g. zinc-phosphate cement why this cement was used if its adhesion to composite materials is very low. In addition, glass-ionomer cements swell in an aqueous environment and in combination with low fracture toughness CAD/CAM composites may increase risk of reconstruction fracture. So, in clinical practice it will be rarely used.

2.        Introduction:

Line 30: polycrystalline ceramics is also used in CAD/CAM technology,

Line 55: GIC is tolerant to moist dentin and enamel,

Line 60: micromechanical interlocking is not caused by self-etching,

Line 61: in chemistry carboxylate is a salt of a carboxylic acid,

Line 64: initiator system doesn’t protect the cement,

Line 68: ….in contrast to cement, RBC luting materials…but RBCs also rank in the cements.

3.        Materials and methods:

Tables with materials composition not included!!

Line 87: why a temporary zinc-phosphate cement was used?

Line 99: how 400 cylinders were prepared from Grandio discs with 98.4 mm in diameter using an Isomet low-speed saw?

Line 113: either static load of 4 kg (kgf? SI units should be used) or a finger pressure applied?

Line 119: ligh intensity units is mW/cm2!!

Line 128: thermally aged in the air or in water bath?

Line 142: if all specimens failed adhesively, why mixed or cohesive fractures are discussed?

Line 144: unclear definition of mixed failures,

Line 164: why did authors say (α=0.05) followed with an alpha risk …5 %??

Line 166: ..correlation between non-parametric material..”???

Line x: no info on Weibull distribution parameters/moduli and their calculation provided.

4.        Results:

Line 173: why interactions between tested variables which are of a second order information is discussed first?

Line x: no info on data normality provided even though a Kolmogorov-Smirnov test was used,

Table 2: Data in two decimal points if the SD is in order of more than 10 %?

Table 3: why a Weibull shape parameter/modulus was used as a criterion of material reliability? Due to ignoring nature of Weibull distribution the modulus in Table 3 was calculated as zero for M and PQM. In fact, this parameter which characterizes variability of data should be very high with these two cements – it is a slope of Weibull plot which was with these systems almost parallel with y-axis reflecting that variability of M and PQM was very small. Unfortunately, close to zero. How was zero value of the modulus calculated? 

Fig 3: brief legend, plot of M strongly affected with one visible outlier (NSB 24 h, material M),

Fig. 4.  Can you explain how tensile and compression zones are defined?

Fig. 5. A) unclear crack, C) subsurface defects are not visible, D) unclear – how this image was obtained?

These are not all criticisms of this manuscript, because there is no point in commenting on others...I hope they will get the authors to reconsider the manuscript.

Author Response

The authors would like to thank the reviewers for taking the time to read and critically appraise the manuscript and for their positive, constructive comments on improving the work.

All comments to the corresponding author have been addressed independently below. The authors’ rebuttal is BLUE, and changes to the revised manuscript in light of the reviewers’ comments are presented in RED.

Comments and Suggestions for Authors

This manuscript could be an interesting study evaluating the bonding efficiency of various types of luting cements to CAD/CAM resin-based composites. However, the study suffers from an inappropriate selection of materials and numerous shortcomings and errors that made me recommend its rejection from such a highly rated scientific journal. I will try to show some of them.

Choice of luting materials - e.g. zinc-phosphate cement why this cement was used if its adhesion to composite materials is very low. In addition, glass-ionomer cements swell in an aqueous environment and in combination with low fracture toughness CAD/CAM composites may increase risk of reconstruction fracture. So, in clinical practice it will be rarely used.

Author’s response: Although it might have been foreseeable that a zinc phosphate cement (or a glass ionomer cement) would not generate sufficient bond strengths when bonded to a resin-based CAD/CAM material, we still share the opinion that it is a valuable part of our research. There are certainly many cases in which the restoration has to be luted temporarily at first or a long-term temporary restoration has been made out of a CAD/CAM RBC. In this context, knowledge of the respective bond strengths is advantageous.

Besides that, these types of cement are still considered as “gold standard” by many practicing dentists, even after a century of use. Some dentists might consider definitive luting a new, unknown restoration material with a zinc phosphate or glass ionomer cement. Hence, the more important it is to be aware of the enormous differences in shear bond strength of a cement compared to a luting composite. There is no other research work that depicts these differences between the luting material categories with regard to CAD/CAM RBC materials as clearly as this study.

  1. Introduction:

Line 30: polycrystalline ceramics is also used in CAD/CAM technology,

Author’s response: We refer to glass ceramics, polymer-infiltrated ceramics and resin-based composites only to name some of the most common restoration materials.

Line 55: GIC is tolerant to moist dentin and enamel,

Author’s response: It is true that GIC is tolerant to the intrinsic moisture of enamel and dentine, but in the initial hardening phase it is also sensitive to excessive moisture (e.g. saliva contamination). We rephrased for proper understanding.

Line 60: micromechanical interlocking is not caused by self-etching,

Author’s response: The respective sentence was rephrased for proper understanding.  

Line 61: in chemistry carboxylate is a salt of a carboxylic acid,

Author’s response: The carboxylate (group) (RCOO-) is meant here.

Line 64: initiator system doesn’t protect the cement,

Author’s response: The resin modification is meant here. It protects the cement against premature exposure to water and dehydration. The sentence was slightly changed to make it more clearly.

Line 68: ….in contrast to cement, RBC luting materials…but RBCs also rank in the cements.

Author’s response: Conventional cements and resin-based composites (RBCs) are different types of luting materials in dentistry. ”Conventional” was added to make this sentence clear.

  1. Materials and methods:

Tables with materials composition not included!!

Author’s response: The compositions of the materials were added to the table.

Line 87: why a temporary zinc-phosphate cement was used?

Author’s response: Please consider the answer above.

Line 99: how 400 cylinders were prepared from Grandio discs with 98.4 mm in diameter using an Isomet low-speed saw?

Author’s response: The manufacturer (Voco GmbH, Cuxhaven, Germany) fabricated several longer cylinders prepared from Grandio discs through CAD/CAM (approximately ø 3 x 18 mm). We cut them into smaller pieces (ø 3 x 2 mm) using a low-speed diamond saw.

Line 113: either static load of 4 kg (kgf? SI units should be used) or a finger pressure applied?

Author’s response: The respective line was corrected (4 kg).

Line 119: ligh intensity units is mW/cm2!!

Author’s response: We apologize for the typo! The light intensity unit was corrected.

Line 128: thermally aged in the air or in water bath?

Author’s response: “Thermally aged in distilled water” was added.

Line 142: if all specimens failed adhesively, why mixed or cohesive fractures are discussed?

Author’s response: All failures occurred in the luting material-CAD/CAM RBC interface. Thus, all obtained values are included in the statistics. These failures were further sub-classified. 

Line 144: unclear definition of mixed failures,

Author’s response: Some parts of luting material bonded on the substrate, others on the cylinder. The luting material parts are complementary and would always result in the original adhesive layer of 7 mm² (100%).

Line 164: why did authors say (α=0.05) followed with an alpha risk …5 %??

Author’s response: This line was corrected.

Line 166: ..correlation between non-parametric material..”???

Author’s response: This line was corrected.

Line x: no info on Weibull distribution parameters/moduli and their calculation provided.

Author’s response: A paragraph on Weibull distribution parameters and their calculation was added.

  1. Results:

Line 173: why interactions between tested variables which are of a second order information is discussed first?

Author’s response: The three- and two-way-ANOVA results are giving the reader a first important overview of our statistical results. To proceed with the statistic, it is important to know if there is an interaction between parameters.

Line x: no info on data normality provided even though a Kolmogorov-Smirnov test was used,

Author’s response: The information was added in 2.4 Statistical analysis.

Table 2: Data in two decimal points if the SD is in order of more than 10 %?

Author’s response: Data in table 2 was changed to one decimal point.

Table 3: why a Weibull shape parameter/modulus was used as a criterion of material reliability? Due to ignoring nature of Weibull distribution the modulus in Table 3 was calculated as zero for M and PQM. In fact, this parameter which characterizes variability of data should be very high with these two cements – it is a slope of Weibull plot which was with these systems almost parallel with y-axis reflecting that variability of M and PQM was very small. Unfortunately, close to zero. How was zero value of the modulus calculated? 

Author’s response: The Weibull statistics are indispensable for strength data and, fortunately, are now also an established tool for dental materials. The logarithm of 0 is not defined and can not be calculated. Please consider that we differentiated between immediately debondend specimens and those who debonded at a later stage of the aging process, which were considered to have a bond strength value different from zero. The Weibull modulus of zero is indeed nonsense - please apologize for the unclear presentation of the data. The respective table and the graphs were corrected for more clarity.

Fig 3: brief legend, plot of M strongly affected with one visible outlier (NSB 24 h, material M),

Author’s response: The visible outlier was a very low bond strength measurement (<0.1 MPa). The table and the legend were adjusted.

Fig. 4.  Can you explain how tensile and compression zones are defined?

Author’s response: Tensile and compression zone were changed to the upper and lower half of the bonded area. Pure shear force was assumed.

Fig. 5. A) unclear crack, C) subsurface defects are not visible, D) unclear – how this image was obtained?

Author’s response: A. Arrows were added in Fig. 5 a to identify the crack. C. white arrow identifies subsurface defect D. Selected specimens were cut perpendicular to the feed direction of the guillotine with a low-speed diamond saw (Isomet low-speed saw), to allow direct sight of the substrate in depth. The cross-sectional area was then finished with SiC abrasive papers (1200 grit, 1500 grit, 2000 grit, 2500 grit and 4000 grit) (Hermes), after which it was polished with a diamond sprayed (DP-Spray, Struers GmbH, Puch, Austria) polishing cloth (DP-Pan 200 mm, Struers GmbH). Before the analysis the specimens had to be sputtered with a 58 nm thick gold-palladium layer (Polaron Range Sputter Coater SC7620, Quorum Technologies, Newhaven, Eng-land). Micrographs were taken using a SEM (Zeiss Supra 55 V P, Carl Zeiss AG) operating at 10 kV (see Materials and Methods).

These are not all criticisms of this manuscript, because there is no point in commenting on others...I hope they will get the authors to reconsider the manuscript.

Reviewer 5 Report

Dear Authors, congratulations for the study and for the data gathered so far. The paper is interesting, however some points have to be settled before the acceptance for publication.

Abstract

Line 17: specifies that the number of total samples: n 400.

Introduction

Reformulate the aim of the study: to add primary endpoint to end of phrase -line 77-:The aim of this study was to evaluate the bonding reliability of 5 luting materials categories to a CAD-CAM RBC;  In addition, I suggest to add “evaluate the strength and incidence of fractures and the index of cohesion with or without sandblasting over time (even in samples subjected to wear)” as secondary endpoints.  Please clarify your aims specifying the outcomes and the endpoints.

Results

Figure 2 (pag. 5 of 13): as well as in the description of the figure, the graphs should also write A materials after 24 hours, B materials after aging, to be easy to read.

Table 2: reformulate the caption and table for easier understanding: for example you might add an asterisk to identify groups that are statistically significantly different.

Author Response

The authors would like to thank the reviewers for taking the time to read and critically appraise the manuscript and for their positive, constructive comments on improving the work.

All comments to the corresponding author have been addressed independently below. The authors’ rebuttal is BLUE, and changes to the revised manuscript in light of the reviewers’ comments are presented in RED.

Comments and Suggestions for Authors

Dear Authors, congratulations for the study and for the data gathered so far. The paper is interesting, however some points have to be settled before the acceptance for publication.

Abstract

Line 17: specifies that the number of total samples: n 400.

Author’s response: The abstract was corrected: Bond strength measurements of the 400 resulting specimens were carried out […].

Introduction

Reformulate the aim of the study: to add primary endpoint to end of phrase -line 77-:The aim of this study was to evaluate the bonding reliability of 5 luting materials categories to a CAD-CAM RBC;  In addition, I suggest to add “evaluate the strength and incidence of fractures and the index of cohesion with or without sandblasting over time (even in samples subjected to wear)” as secondary endpoints.  Please clarify your aims specifying the outcomes and the endpoints.

Author’s response: The aims of the study were reformulated: Given the wide variety of luting materials available, the aim of this study was to evaluate the bonding strength and reliability of five luting material categories to a CAD/CAM RBC. In addition, the incidence of fractures and the index of cohesion with or without sandblasting over time are considered. 

Results

Figure 2 (pag. 5 of 13): as well as in the description of the figure, the graphs should also write A materials after 24 hours, B materials after aging, to be easy to read.

Author’s response: Figure 2 was corrected.

Table 2: reformulate the caption and table for easier understanding: for example you might add an asterisk to identify groups that are statistically significantly different.

Author’s response: In Table 2 uppercase letters refer to within-row comparisons, while lowercase letters refer to within-column comparisons. Identical letters identify groups that are not statistically significantly different. Adding asterisks instead of letters wouldn’t have been sufficient, because the groups were compared in rows and columns.

Round 2

Reviewer 1 Report

The authors addressed the comments made by the reviewers, I appreciate it. However, to my understanding, I do not see the real novelty of this study. Overall, I would recommend to reformulate the introduce and discussion of the study to match the content with the JFB profile.

Line 138: Please explain the use of distilled water in research, not artificial oral fluid.

Lines 144 and 157: The diagrams presented in Figs. 1 and 2 should be dimensioned.

In section Results: It is worth adding shear curves.

Boiled water is not distilled water, boiling alone will not remove mineral salts, e.g. calcium, and this is important for storing dental materials.

Author Response

All comments to the corresponding author have been addressed independently below. The authors’ rebuttal is BLUE, and changes to the revised manuscript in light of the reviewers’ comments are presented in RED.

Comments and Suggestions for Authors

The authors addressed the comments made by the reviewers, I appreciate it. However, to my understanding, I do not see the real novelty of this study. Overall, I would recommend to reformulate the introduce and discussion of the study to match the content with the JFB profile.

Line 138: Please explain the use of distilled water in research, not artificial oral fluid.

Author’s response: We understand the concern and have explained in detail our decision to use dest water for storage. The following paragraph was added to the discussion:

"All samples in this study were stored and aged in distilled water. This decision was made due to the fact that distilled water is the medium described in all standards for testing the bond strength and mechanical properties of dental materials. Since significantly more ions were leached from resin-based composite fillers in artificial saliva than in distilled water [31], greater decrease in adhesive strength would be expected with aging in artificial saliva. However, a recent study investigating the influence of aging in artificial saliva compared to distilled water on the bond strength of resin-based composites demonstrated the opposite [32]. Furthermore, the decision for the choice of medium also concerns the fact that the mechanical properties of resin-based composites [33] and glass ionomer cements [34] were not affected by the storage medium, which was either distilled water or artificial saliva".

Lines 144 and 157: The diagrams presented in Figs. 1 and 2 should be dimensioned.

Author’s response: We corrected accordingly.

In section Results: It is worth adding shear curves.

Author’s response: We believe that adding a shear curve to the manuscript is too elementary, as it is basic row data. Shear curves represent the applied force in [N] and the corresponding deformation in % for every single specimen, which would result in 400 shear curves for the present study. Thus, the presentation of raw data in the manuscript (like shear curves) is generally neither necessary nor recommended.

Boiled water is not distilled water, boiling alone will not remove mineral salts, e.g. calcium, and this is important for storing dental materials.

Author’s response: The distilled water used in this study was produced by H. Kerndl GmbH, Weißenfeld, Germany. It was demineralized through reversed osmosis.